# Immunological Effects of *Aster yomena* Callus-Derived Extracellular Vesicles as Potential Therapeutic Agents against Allergic Asthma

**DOI:** 10.3390/cells11182805

**Published:** 2022-09-08

**Authors:** Woo Sik Kim, Ji-Hye Ha, Seong-Hun Jeong, Jae-In Lee, Ba-Wool Lee, Yu Jeong Jeong, Cha Young Kim, Ji-Young Park, Young Bae Ryu, Hyung-Jun Kwon, In-Chul Lee

**Affiliations:** 1Functional Biomaterial Research Center, Korea Research Institute of Bioscience and Biotechnology (KRIBB), Jeongeup 56212, Korea; 2College of Veterinary Medicine and BK21 FOUR Program, Chungnam National University, Daejeon 34131, Korea; 3Laboratory of Veterinary Pathology, College of Veterinary Medicine, Chonnam National University, Gwangju 61186, Korea; 4Division of Biotechnology and Advanced Institute of Environment and Bioscience, Jeonbuk National University, Iksan 54596, Korea; 5College of Verinary Medicine and BK21 FOUR Program, Chonnam National University, Gwangju 61186, Korea; 6Biological Resource Center, Korea Research Institute of Bioscience and Biotechnology (KRIBB), Jeongeup 56212, Korea

**Keywords:** extracellular vesicles, *Aster yomena*, callus, dendritic cells, T cell-mediated inflammatory diseases, asthma, T cells

## Abstract

Plant-derived extracellular vesicles, (EVs), have recently gained attention as potential therapeutic candidates. However, the varying properties of plants that are dependent on their growth conditions, and the unsustainable production of plant-derived EVs hinder drug development. Herein, we analyzed the secondary metabolites of *Aster yomena* callus-derived EVs (AYC-EVs) obtained via plant tissue cultures and performed an immune functional assay to assess the potential therapeutic effects of AYC-EVs against inflammatory diseases. AYC-EVs, approximately 225 nm in size, were isolated using tangential flow filtration (TFF) and cushioned ultracentrifugation. Metabolomic analysis, using ultra-performance liquid chromatography-quadrupole time-of-flight mass spectrometry (UPLC-QTOF/MS), revealed that AYC-EVs contained 17 major metabolites. AYC-EVs inhibited the phenotypic and functional maturation of LPS-treated dendritic cells (DCs). Furthermore, LPS-treated DCs exposed to AYC-EVs showed decreased immunostimulatory capacity during induction of CD4^+^ and CD8^+^ T-cell proliferation and activation. AYC-EVs inhibited T-cell reactions associated with the etiology of asthma in asthmatic mouse models and improved various symptoms of asthma. This regulatory effect of AYC-EVs resembled that of dexamethasone, which is currently used to treat inflammatory diseases. These results provide a foundation for the development of plant-derived therapeutic agents for the treatment of various inflammatory diseases, as well as providing an insight into the possible mechanisms of action of AYC-EVs.

## 1. Introduction

Natural bioactive substances, produced by various organisms, have been extensively studied for various reasons over the past decades [1,2,3]. Extracellular vesicles (EVs), secreted by cells for information exchange, have received increasing attention owing to several of their properties, such as a wide range of functionalities, biocompatibility, low immunogenicity, and target specificity, for preclinical and clinical applications [4,5].

EVs, approximately 30–1000 nm in size, are involved in the intercellular transport of nucleic acids, proteins, lipids, and metabolites by encapsulating these within EV lipid bilayers [5]. Importantly, the encapsulated materials and functionality of EVs can vary according to the origin, characteristics, and conditions of the cells, as well as environmental changes [6,7,8]. Owing to this characteristic, EVs derived from prokaryotic and eukaryotic cells have wide applications in the areas of research, diagnostics, and therapy for various diseases. EVs isolated from living organisms can be used to control or treat various diseases. For instance, clinical studies have investigated the potential of mesenchymal stem cell-derived EVs in the treatment of immune disorders, cardiovascular diseases, neurological diseases, wounds, and COVID-19 [4,9]. In addition, various studies have demonstrated that EVs isolated from probiotic strains of microorganisms, such as *Lactobacillus paracasei* and *Propionibacterium freudenreichii*, can regulate excessive inflammatory reactions [10,11]. Thus, research on the biological, functional, and immunological characteristics of EVs derived from various organisms may lead to advances in the food, pharmaceutical, and medical industries. 

Similar to animal-derived EVs, plant-derived EVs act as protective vessels that are involved in transporting various substances between cells and contribute to plant growth and development, protective responses, and plant–microbe coexistence [12,13]. Recent studies have demonstrated that EVs isolated from grapes, ginger, broccoli, and grapefruits perform antioxidant and anti-inflammatory activities inside their cells [14,15]. Despite the benefits of plant-derived EVs, a number of drawbacks exist because the functional characteristics of plant-derived EVs can vary depending on the growth conditions [16]. Nevertheless, due to the advent of plant tissue culture, a valuable technique, this particular problem can be overcome. Plant tissue culture can also be used for conserving and restoring endangered plants, as well as growing a complete plant from its constituent parts (cells, tissues, organs, embryo, etc.) under sterile conditions [17,18]. The technique generally comprises two steps: induction of dedifferentiated calli from plant tissues and redifferentiation of a callus capable of doing so [17]. Many bioactive substances produced by the mother plant can be produced using plant callus culture [19]. Paclitaxel (brand name: CyviloqTM), developed by Samyang Genex, South Korea, is an example of a medication developed using callus cultures [20,21]. 

Recently, we established a callus culture system for *Aster yomena* (*A. yomena*) and demonstrated the various bioactivities of substances derived from *A. yomena* callus, such as anti-photoaging, anti-melanogenic, and anti-inflammatory effects [22,23]. We found that the *A. yomena* callus culture system could induce EV formation, resembling the induction of EV formation in other plant species, and that these EVs contained secondary metabolites with various immunological effects. We examined the immune functional characteristics of *A. yomena* callus-derived EVs (AYC-EVs) to verify the feasibility of using them as agents for the treatment or alleviation of diseases. We also verified the therapeutic potential of AYC-EVs against allergic asthma in animal models of asthma. 

## 2. Materials and Methods

### 2.1. Isolation and Characterization of Extracellular Vesicles in Aster yomena Callus Culture Supernatant

The culture methods for *A. yomena* calli have been described in our previous study [22]. The callus used in this study was established from an *A. yomena* root. Briefly, surface-sterilized root segments (approximately 1 cm) from *A. yomena* were incubated on agar-solidified MS1D medium (Sigma-Aldrich, St. Louis, MO, USA) in the presence of growth regulators (1 mg/L 2,4-dichlorophenoxyacetic acid (2,4-D), 3% (*w*/*v*) sucrose, and 0.4% (*w*/*v*) Gelrit) at 25 °C in the dark. At 2 weeks after incubation, calli that were formed from *A. yomena* roots were collected, and a 500-mL flask containing 200 mL of liquid MS1D medium was inoculated with 20 g of callus. The inoculated calli were cultured on a rotary shaker (80 rpm) at 25 °C in the dark. At 2 weeks after incubation, AYC-EVs were isolated from culture supernatants. Non-EVs, including cells, cell debris, microparticles, and apoptotic bodies, were removed using the following process. First, callus pellets in the culture supernatant were depleted by centrifugation at 1800× *g* for 20 min at 4 °C, and the supernatant fluids were centrifuged again at 3500× *g* for 20 min at 4 °C. The harvested supernatants were centrifuged at 10,000× *g* for 20 min at 4 °C and the pellets were depleted once again. The supernatant fluids that were depleted of non-EVs were filtered through a 0.2-µm vacuum filter (ThermoFisher Scientific, Waltham, MA, USA) to remove potential bacterial contaminants. Next, EVs in the filtered supernatants were isolated using a combination of tangential flow filtration (TFF; Pall Life Sciences, Port Washington, NY, USA) and cushioned ultracentrifugation. In the size-based particle separation method, filtered supernatants (1 L) were concentrated to 35 mL using TFF with a 500-kDa MWCO ultrafiltration membrane filter capsule (Pall Life Sciences). Next, supernatants (35 mL) mixed with soluble proteins and AYC-EVs were added to an ultracentrifuge tube (38.5 mL; Beckman Coulter, California, USA) for the SW 32 Ti Rotor (Beckman Coulter) and 3 mL Lymphoprep^TM^ solutions (density 1.077 g/mL; StemCell Technologies, Vancouver, BC, Canada), as cushioning buffer, were loaded using a Pasteur pipette (Sigma-Aldrich) into a tube containing 35 mL of supernatant. The tubes were centrifuged at 120,000× *g* for 60 min at 4 °C using a Beckman Coulter ultracentrifuge (Opima XE-100). After ultracentrifugation, a volume of 4 mL from the bottom of the ultracentrifuge tubes consisting of 3 mL cushioning buffer and 1 mL supernatant (including AYC-EVs) was carefully collected and diluted to 100 mL with cold PBS as the dialysis buffer. A second TFF (at 500-kDa cut-off) process was then performed to concentrate the EVs and change the buffer. A total volume of 104 mL (consisting of 3 mL of cushioning buffer, 1 mL of AYC-EVs, and 100 mL of PBS) was concentrated to 5 mL to obtain AYC-EVs. Finally, 5 mL of AYC-EVs were filtered through a 0.22-μm syringe filter (Thermo Fisher Scientific) for use in in vitro and in vivo experiments and stored at −80 °C until further use. 

The protein concentrations of AYC-EVs were analyzed using a BCA protein assay kit (Thermo Fisher Scientific). The size distribution of AYC-EVs was characterized using transmission electron microscopy (TEM) and dynamic light scattering (DLS). For TEM analysis, 5 μL of AYC-EVs were dropped onto a formvar-carbon-coated TEM grid (Electron Microscopy Sciences, Hatfield, PA, USA) and negatively stained using 2% uranyl acetate (Sigma-Aldrich, St. Louis, MO, USA). Images were visualized using a TEM (JEOL-2100F, Tokyo, Japan). In addition, the size of the AYC-EVs was evaluated using DLS (Malvern Instruments, Malvern, UK), according to the manufacturer’s protocol.

### 2.2. Isolation of Aster yomena Callus Extracts

The inoculated calli were cultured on a rotary shaker (80 rpm) at 25 °C in the dark, as described in the *Materials and Methods* section, for AYC-EV isolation. At 2 weeks after incubation, callus pellets were collected to isolate the pellet extracts (AYC-P-E) of *A. yomena* calli via centrifugation at 1800× *g* for 20 min at 4 °C. The supernatant fluids (AYC-S-E) were freeze-dried and resuspended in 5 mL of PBS. Then, 5 mL of AYC-S-E was filtered through a 0.22 μm syringe filter and used for in vitro experiments. The pellets were suspended in 500 mL of deionized water and incubated at 80 °C for 2 h. Next, incubated samples were centrifuged at 10,000× *g* for 20 min at 4 °C, and supernatants were freeze-dried and resuspended in 5 mL of PBS. Finally, 5 mL of AYC-P-E was filtered through a 0.22 μm syringe filter and used for in vitro experiments.

### 2.3. Characterization and Identification of the Major Metabolites in AYC-EVs

Characterization and identification of major metabolites present in AYC-EVs was carried out using ultra-performance liquid chromatography-quadrupole time-of-flight mass spectrometry (UPLC-QTOF/MS). AYC-EVs powders (1 mg) were first produced using a vacuum freeze dryer (VD-800F; Taitec, Saitama-ken, Japan) and dissolved in 80% methanol (100 μL). Subsequently, AYC-EV solutions (10 μL) dissolved in methanol were injected into an Acquity UPLC BEH C18 column (Waters, Milford, MA, USA) that was equilibrated with ultrapure water supplemented with 0.1% formic acid. Then, the column was eluted using a gradient with acetonitrile, supplemented with 0.1% formic acid, under a flow rate of 0.35 mL/min (total 9 min). The eluted metabolites were measured using QTOF/MS in positive electrospray ionization (ESI) mode as follows: capillary voltage: 3 kV, sampling cones: 30 V, source temperature: 100 °C, desolvation flow rate: 800 L/h, and sheath gas temperature: 400 °C. TOF/MS and MS/MS scan ranges were *m/z* 50–1500 and *m/z* 50–1000, respectively. Leucine-enkephalin at *m/z* 556.2771 for positive mode was used as the lock mass (10 s frequency). UNIFI software (Waters) was used for collection, normalization, and alignment of the MS datasets. The identification of metabolites was obtained by using web-based bioinformatics databases (Human Metabolome Database, Metabolite and Chemical Entity database, and Chemspider database, each accessed on January 3 2022), and authentic reference standards.

### 2.4. Differentiation of Dendritic Cells (DCs) from Mouse Bone Marrow

Bone marrow cells were isolated from 8-week-old female C57BL/6 mice (Orient Bio, Seongnam, South Korea) and cultured in an incubator at 37 °C and 5% CO_2_ in complete RPMI1640 medium (Gibco BRL, Grand Island, NY, USA) containing 20 ng/mL GM-CSF (JW Creagene, Gyeonggi, Korea), 2.5 ng/mL IL-4 (JW Creagene), 10% fetal bovine serum (Gibco BRL), and 1% antibiotics (penicillin/streptomycin, Gibco BRL). On the 8th day, the cells were harvested and labeled with CD11c MicroBeads (Miltenyi Biotec, San Diego, CA, USA) at 4 °C for 15 min. After incubation, CD11c-labeled cells (CD11c^+^ cells) were isolated using MACS LS separation columns (Miltenyi Biotec), according to the manufacturer’s protocol. The purity of the CD11c^+^ dendritic cells (DCs) was determined to be >90%.

### 2.5. Analysis of DC Viability and Cytotoxicity of AYC-EVs in DCs 

To analyze the viability of immature and mature DCs, DCs (0.5 × 10^6^ cells per well in 500 μL complete RPMI1640 media) were incubated with or without 100 ng/mL lipopolysaccharide (LPS; Invitrogen, San Diego, CA, USA) as a positive control for DC maturation for 2 h, and then treated with various concentrations (1–50 μg/mL) of AYC-EVs for 24 h. After stimulation, cell viability was analyzed using the EZ-Cytox Cell Viability Assay Kit (DoGen, Seoul, Korea), according to the manufacturer’s protocol. Briefly, 50 μL of EZ-Cytox kit reagent was added to each well and incubated at 37 °C for 60 min. After incubation, absorbance at 450 nm was measured using a microplate reader (Molecular Devices Inc., San Jose, CA, USA). For cytotoxicity analysis, cells (non-, LPS-, AYC-EV-, AYC-EV, and LPS-treated DCs) were harvested 24 h after AYC-EV treatment and stained with Annexin V and propidium iodide (PI) using FITC Annexin V/Dead Cell Apoptosis Kit (Thermo Fisher Scientific), according to the manufacturer’s protocol. Annexin V/PI-stained cells were analyzed using a Life Launch Attune Nxt Flow Cytometer (Thermo Fisher Scientific) and FlowJo software (Version 10; Tree Star, Inc., Ashland, OR, USA). 

### 2.6. Analysis of Extracellular and Intracellular Cytokines in DCs

To analyze extracellular cytokines, DCs were incubated with or without 100 ng/mL LPS for 2 h and then treated with AYC-EVs (2, 10, and 20 μg/mL) for 24 h. The culture supernatants were harvested 24 h after AYC-EV treatment, and extracellular cytokine levels for TNF-α, IL-12p70, and IL-10 were analyzed using a cytokine-specific sandwich ELISA kit (ThermoFisher Scientific), according to the manufacturer’s protocol. To analyze intracellular cytokines, DCs were incubated with 100 ng/mL LPS for 2 h and treated with AYC-EVs (2, 10, and 20 μg/mL) for 12 h in the presence of 1x protein transport inhibitor cocktail (Thermo Fisher Scientific). Subsequently, the cells were harvested and washed twice with cold PBS. Next, the cells were stained with anti-CD11c antibody (PE-Cy7; BD Bioscience, San Jose, CA, USA) and Live/Dead Fixable Dead Cell Stain kit (L/D; Invitrogen) reagent for 15 min at room temperature and then fixed and permeabilized using a BD Cytofix/Cytoperm kit reagent, according to the manufacturer’s protocol. Finally, the cells were washed twice with BD Perm/Wash Buffer and stained with anti-TNF-α (APC; Thermo Fisher Scientific), anti-IL-12p70 (FITC; BD Bioscience), and anti-IL-10 (PE; BD Biosciences) antibodies for 20 min at room temperature. Fluorescein-stained cells were analyzed using a Life Launch Attune Nxt flow cytometer.

### 2.7. Analysis of Immune-Modifying Surface Molecules on DCs

To evaluate whether the expression of surface molecules on immature and mature DCs could be regulated through AYC-EV treatment, DCs were incubated with or without 100 ng/mL LPS for 2 h, and then treated with AYC-EVs (2, 10, and 20 μg/mL). At 24 h after AYC-EVs treatment, the cells were harvested and washed twice with cold PBS. Next, cells were stained with anti-CD11c (PE-Cy7), anti-CD80 (APC; BD Biosciences), anti-CD86 (FITC; BD Biosciences), anti-MHC-I (PE; BD Biosciences), and anti-MHC-II antibodies (PerCp-Cy5.5; BD Biosciences) and L/D reagent for 15 min at room temperature. CD80-, CD86-, MHC-I-, and MHC-II-positive CD11c^+^ cells were analyzed using a Life Launch Attune Nxt Flow Cytometer. All isotype controls (APC-conjugated Hamster IgG2, FITC-conjugated rat IgG2a, PE-conjugated rat IgG2a, and PerCp-Cy5.5-conjugated rat IgG2b) for surface molecules were purchased from BD Biosciences. 

### 2.8. Analysis of Antigen Uptake and Antigen Presenting Ability on DCs

To analyze their antigen uptake ability, DCs were incubated with 100 ng/mL LPS for 2 h and then treated with AYC-EVs (2, 10, and 20 μg/mL) for 24 h. After treatment, 0.5 mg/mL FITC-conjugated Dextran (40,000 Da, Sigma-Aldrich) was added to each well and then incubated at 37 °C or 4 °C for 30 min. Cell culture conditions at 4 °C were used as negative controls for those at 37 °C. Next, the cells were harvested and washed three times with cold PBS. CD11c^+^ dextran ^+^ cells were analyzed using a Life Launch Attune Nxt flow cytometer. For analysis of antigen presenting ability, antigen presentation induced by MHC-I was examined, based on treatment with ovalbumin (OVA) protein (Sigma-Aldrich) or OVA_257–264_ peptide (SIINFEKL; positive control for MHC-I-antigen presentation) in each experimental condition. In addition, antigen presentation induced by MHC-II was examined upon treatment with Eα_44–76_ peptide (RLEEFAKFASFEAQGALANIAVDKANLDVMKKR; underlined sequence bound to MHC-II) or Eα_52–68_ peptide (ASFEAQGALANIAVDKA; positive control for MHC-II-antigen presentation). Briefly, DCs were incubated with 100 ng/mL LPS for 2 h and then treated with AYC-EVs (20 μg/mL) for 24 h in the presence of OVA (500 μg/mL) or Eα_44–76_ peptide (25 μg/mL). The cells were harvested and washed twice with cold PBS. Next, the cells were stained with anti-CD11c (PE-Cy7) and Y-Ae (PE, Thermo Fisher Scientific) or I-Ab antibodies (FITC, Thermo Fisher Scientific). Here, the H-2Kb antibody was expected to react with the MHC-I self OVA_257–264_ peptide and the Y-Ae antibody was expected to react with the MHC-II self Eα_52–68_ peptide. CD11c^+^H-2Kb^+^ and CD11c^+^Y-Ae^+^ cells were analyzed using a Life Launch Attune Nxt flow cytometer.

### 2.9. Allogenic Mixed Lymphocyte Reaction Assay

An allogeneic mixed lymphocyte reaction (MLR) assay was conducted using DCs from an 8-week-old female C57BL/6 mouse and T cells from an 8-week-old female BALB/c mouse. First, DCs were incubated with 100 ng/mL LPS for 2 h and then treated with AYC-EVs (20 μg/mL) for 24 h. These cells were then used in the MLR assay. Spleen cells were isolated from the BALB/c mouse and labeled with CD4 or CD8 microbeads (Miltenyi Biotec) at 4 °C for 15 min. After incubation, CD4- or CD8-labeled cells were isolated using MACS LS separation columns, according to the manufacturer’s protocol. Isolated T cells were stained using the CellTrace CFSE Cell proliferation kit reagent (2 μM, Invitrogen) for 10 min in a 37 °C water bath. Next, 96-well plates were coated with anti-CD3e (1 μg/mL; Invitrogen) antibody for 2 h at 37 °C, and the cells (DCs; 2 × 10^5^ cells per well and CFSE-labeled T cells; 1 × 10^6^ cells per well) were co-cultured with anti-CD28 antibody (1 μg/mL; Invitrogen) in anti-CD3e-coated plates. After 2 days of co-culture, cell culture supernatants were harvested, and IFN-γ, TNF-α, IL-2, IL-5, and IL-17A levels were analyzed using a cytokine-specific ELISA kit. Kits were purchased from Thermo Fisher Scientific. Finally, the harvested T cells were stained with anti-CD4 or anti-CD8 antibodies, and T cell proliferation levels were analyzed using a Life Launch Attune Nxt Flow Cytometer.

### 2.10. Therapeutic Efficacy of AYC-EVs in a Mouse Model of Ovalbumin-Mediated Allergic Asthma

The therapeutic efficacy of AYC-EVs in a mouse model of ovalbumin (OVA)-mediated allergic asthma was evaluated in compliance with the Institutional Animal Care and Use Committee (permit number: KRIBB-AEC-21120), with approval from the Korea Research Institute of Bioscience and Biotechnology (KRIBB). Herein 10-week-old female BALB/c mice (weight:20 ± 2 g) were sourced from Orient Bio and housed at the KRIBB animal facility under specific pathogen-free conditions at constant temperature (22 ± 2 °C), light/dark cycle (12 h/12 h), and humidity (55 ± 5%). In addition, the mice were acclimated for 1 week after transport to a new facility and randomly divided as follows (n = 5 per group): control group (Group 1), OVA group (Group 2), OVA/DEX group (Group 3), and OVA/AYC-EV group (Group 4). For the control groups, mice were injected intraperitoneally with 200 μL of PBS on days 1 and 8. Next, the PBS-injected mice were orally administered 200 μL PBS once per day on days 12–17. Finally, the mice were challenged for 1 h with PBS via an ultrasonic nebulizer (NE-U12; OMRON Corp., Tokyo, Japan) once per day on days 15, 16, and 17. For the OVA groups, mice were sensitized intraperitoneally with 200 μL of PBS containing OVA (40 μg, Sigma-Aldrich) and aluminum hydroxide gel (2 mg, Sigma-Aldrich) on days 1 and 8. Next, OVA-injected mice were orally administered 200 μL of PBS once per day on days 12–17. Finally, the mice were challenged for 1 h with OVA (1%, *w*/*v*, in PBS) once per day via a nebulizer on days 15, 16, and 17. For the OVA/DEX groups, mice were sensitized intraperitoneally with 200 μL of PBS containing OVA (40 μg) and aluminum hydroxide gel (2 mg) on days 1 and 8. Next, OVA-injected mice were orally administered dexamethasone (3 mg/kg, Sigma-Aldrich) once per day on days 12–17. Finally, the mice were challenged for 1 h with OVA (1%, *w*/*v*, in PBS) once per day via a nebulizer on days 15, 16, and 17. For the OVA/AYC-EV groups, mice were sensitized intraperitoneally with 200 μL of PBS containing OVA (40 μg) and aluminum hydroxide gel (2 mg) on days 1 and 8. Next, OVA-injected mice were orally administered AYC-EVs (4 and 8 mg/kg) once per day, on days 12–17. Finally, the mice were challenged for 1 h with OVA (1%, *w*/*v*, in PBS) once per day via a nebulizer on days 15, 16, and 17. All groups were euthanized after 19 days. 

### 2.11. Analysis of Airway Resistance in a Mouse Model of Allergic Asthma Induced by Ovalbumin Antigen

At 24 h after the last OVA challenge, airway resistance values were assessed via Flexivent (SCIREQ Scientific Respiratory Equipment Inc., Montreal, PQ, Canada) [24]. Mice were anesthetized with an intraperitoneal injection of avertin (250 mg/kg, Sigma-Aldrich). After baseline impedance measurements, the mice were placed in a chamber and nebulized with aerosolized PBS or methacholine (10 and 40 mg/mL) via an Aeroneb nebulizer (SCIREQ) for 10 s. Airway resistance levels were measured every 30 s for 1 min and then refreshed for 2 min. The volume history of the lung was established with 6-s deep inflations to a pressure limit of 30 cm H_2_O.

### 2.12. Analysis of ELISA and Inflammatory Cells in Bronchoalveolar Lavage Fluid (BALF)

BALF samples were obtained in accordance with a previously described method [25]. Mice were sacrificed at 48 h after last OVA challenge under isoflurane inhalation anesthesia, and a tracheostomy was conducted. To obtain BALF, ice-cold PBS (0.7 mL) was infused into the lungs twice and withdrawn each time using a tracheal cannula (total volume of 1.4 mL). The BALF samples were centrifuged at 1000 rpm for 10 min at 4 °C, and the collected supernatants were stored at –70 °C. The collected pellets were resuspended in 250 μL PBS and centrifuged to attach the cells to a slide (1000 rpm for 5 min at 20 °C) using a Cytospin 4 centrifuge (Thermo Scientific, Waltham, MA, USA). For differential cell count, slides were dried, and cells were fixed and stained with Diff-Quik^®^ staining reagent (IMEB, San Marcos, CA, USA) according to the manufacturer’s instructions. Five images of each slide were captured using a Leica DM5000B microscope and Leica Application Suite acquisition software (Leica Microsystems, Wetzlar, Germany) under a 40× objective lens. Next, the total cells, eosinophils, antigen-presenting cells (APCs, including DCs and macrophages), neutrophils, and lymphocytes were enumerated. TNF-α (R&D Systems, Minneapolis, MN, USA), IL-4 (R&D Systems), IL-5 (R&D Systems), IL-13 (R&D Systems), eotaxin (R&D Systems), and mucin 5AC (MUC5AC; Cusabio Biotech Co., Wuhan, China) levels in supernatants were measured with each ELISA kit, following the manufacturer’s instructions. The level of total IgE and OVA-specific IgE in the serum were measured using ELISA kits (BioLegend, CA, USA), following the manufacturer’s instructions. Absorbance was measured at 450 nm using a microplate reader (iMarkTM; Bio-Rad Laboratories, Richmond, CA, USA).

### 2.13. Histopathology

After collecting BALF, the dissected lung tissues were fixed in 10% (*v*/*v*) neutral-buffered formalin for 48 h at room temperature. Fixed tissues were embedded in paraffin, and cut into 4-μm-thick sections. The slides were stained with hematoxylin and eosin (H&E) (BBC Biochemical, Mount Vernon, WA, USA) or periodic acid-Schiff (PAS), respectively, to evaluate the extent of airway inflammation or mucus production. The quantitative determination for airway inflammation or mucus production was performed and the degree of each lesion was graded in a blinded manner using a light microscope (Leica Microsystem, Wetzlar, Germany) with 10× and 20× objective lenses, on the following scale: 0, no lesions; 1, minimal; 2, mild; 3, moderate; and 4, severe. 

### 2.14. Analysis of Splenic DC and CD4^+^ T Cell Responses in a Mouse Model of Ovalbumin-Mediated Allergic Asthma

After mice were euthanized, splenocytes from each group were isolated and stained with L/D reagent, lineage marker cocktails (FITC-conjugated anti-CD3, anti-CD19, and anti-NK1.1 antibodies; BD Biosciences) and anti-CD11c (PE-Cy7), anti-CD80 (APC), anti-MHC-I (PE), and MHC-II (PerCp-Cy5.5) antibodies for 15 min at room temperature. After staining, splenic DC (lineage^-^CD11c^+^MHC-II^+^) maturation (CD80 and MHC-I expression on lineage^-^CD11c^+^MHC-II^+^ cells) was analyzed using a Life Launch Attune Nxt Flow Cytometer. For the analysis of Th1, Th2, Th17, and Th2/Th17 cells, splenocytes from each group were isolated and stimulated with 1× Cell Stimulation Cocktail (Thermo Fisher Scientific) supplemented with PMA, ionomycin, and protein inhibitors for 4 h at 37 °C. For analysis of Th1, Th2, and Th17 cells, cells were harvested and stained with anti-CD3 antibody (Alexa 700; BD Biosciences), anti-CD4 antibody (PerCp-Cy5.5; BD Bioscience), and L/D reagent for 15 min at room temperature, and then fixed and permeabilized using BD Cytofix/Cytoperm kit reagents according to the manufacturer’s protocol. Cells were then washed with BD Perm/Wash Buffer and stained with anti-IFN-γ (PE; Thermo Fisher Scientific), anti-IL-5 (APC; BD Bioscience), and anit-IL-17A (PE-Cy7; Thermo Fisher Scientific) antibodies for 20 min at room temperature. For analysis of regulatory T cells, splenocytes from each group were directly stained with anti-CD3 antibody (Alexa 700), anti-CD4 antibody (PerCp-Cy5.5), anti-CD25 antibody (APC; ThermoFisher Scientific), and L/D reagent for 15 min at room temperature. Cells were washed with cold PBS, fixed, and permeabilized using Foxp3/Transcription Factor Staining Buffer Set (Thermo Fisher Scientific) reagents according to the manufacturer’s protocol. After several washes with Fixation/Per Dilunt (Thermo Fisher Scientific), cells were stained with anti-Foxp3 (PE; Thermo Fisher Scientific) antibody for 20 min at room temperature. After staining, regulatory T cells (CD3^+^CD4^+^CD25^+^Foxp3^+^) were analyzed using a Life Launch Attune Nxt Flow Cytometer. The absolute number of splenic DCs and T cell subsets was analyzed by multiplying the percentage of each cell type by the total cell number in the single-cell suspension of each group.

### 2.15. Statistical Analysis 

Statistical differences for all data were assessed using GraphPad Prism 7.0 software (San Diego, CA, USA). Differences between groups (two or three groups) were evaluated using unpaired *t*-tests and one-way ANOVA, respectively. The following *p*-value thresholds of significance were used: * *p* < 0.05, ** *p* < 0.01, and *** *p* < 0.001.

## 3. Results

### 3.1. Characterization and Metabolite Composition of AYC-EVs

In previous studies, we demonstrated the antioxidant, anti-inflammatory, anti-photoaging, anti-melanogenesis, and anti-wrinkle properties of *A. yomena* callus extracts [22,23]. Therefore, we expected that EVs, which can regulate the host immune system, could be obtained using the plant callus culture system. We first isolated AYC-EVs from the culture supernatant of *A. yomena* calli using the TFF system and cushioned ultracentrifugation, as described in the *Materials and Methods* section (Figure 1a). The size and structure of AYC-EVs were confirmed through DLS (Figure 1b) and TEM (Figure 1c). Figure 1b shows that the size of the AYC-EVs was approximately 225.2 nm, as analyzed using DLS. Next, we investigated the secondary metabolite composition of AYC-EVs, because plant-derived EVs contain a myriad of bioactive molecules (nucleic acids, proteins, lipids, and metabolites) that can have a significant impact on recipient cells [12]. We analyzed the metabolites of AYC-EVs using liquid chromatography/mass spectrometry (LC/MS). A total of 17 metabolites were detected in AYC-EVs (Figure 1d and Table 1). Here, one purine derivative (guanine) and one purine nucleoside (adenosine) were used to form nucleotides of the nucleic acids, and 2 amino acids (L-phenylalanine, L-tryptophan) that make up proteins were detected. Moreover, we identified 13 lipids, including fatty acids, fatty amides, and glycerophospholipids. The detailed results for the 13 lipids were as follows: 5 fatty acyl including omega-3 (α-Linolenic acid), omega-6 (methyl linoleate) fatty acids, and derivates thereof (omega-3 derivatives; 2-hydrolinolenic acid, stearidonic acid, omega-6 derivative; 2-hydrolinolenic acid), 5 fatty amides (belonging to the fatty acyls) including omega-3 derivatives (adrenoyl ethanolamide, octadecanmide, 13-docosenamide), omega-7 derivative (palmitic amide), linoleoyl ethanolamide, and 3 glycerophospholipids, including LysoPC(18:2), LysoPC(16:1), and LysoPC(18:1). As a result, the metabolites included in AYC-EVs were expected to provide important insights into their structure and immunological functions.

### 3.2. AYC-EVs Suppress the Expression of Pro- and Anti-Inflammatory Cytokines Secreted by LPS-Treated DCs

DCs play vital roles in the innate and adaptive immune systems, particularly in relation to protective immunity and immune homeostasis against various diseases mediated by tumors, excessive inflammation, and viral and bacterial infections [26]. Thus, based on the understanding of their roles, DCs can be used as powerful tools for developing new drugs and understanding the pathogenesis of various diseases. In the present study, we aimed to understand the immunological function of AYC-EVs by determining whether treatment with AYC-EVs could induce phenotypic and functional alterations in immature or mature DCs. Prior to confirming the immunological functions of AYC-EVs, we investigated whether AYC-EVs could induce cellular cytotoxicity in immature and mature DCs. When immature (untreated) and mature (LPS-treated) DCs were treated with AYC-EVs, we detected no significant changes in the viability and cytotoxicity of DCs (immature and mature) at various concentrations (1–50 μg/mL) (Figure 2a,b). Next, we evaluated the secretion level and expression pattern of pro- and anti-inflammatory cytokines induced by non- and LPS-treated DCs exposed to non-toxic concentrations of AYC-EVs, as these cytokines play important roles in initiating and controlling T cell immunity [27,28]. We found that treatment with AYC-EVs significantly decreased the production of anti (IL-10)- and pro (TNF-α, IL-12p70)-inflammatory cytokines secreted by mature DCs in a dose-dependent manner (Figure 2c; extracellular cytokine levels, and d; intracellular cytokine levels). However, a change in cytokine expression pattern in immature DCs was not observed after treatment with AYC-EVs (Figure 2c). Additionally, AYC-EVs were more effective at reducing the production of pro-inflammatory cytokines (TNF-α, IL-12p70) in LPS-treated DCs than in AYC-P-E and AYC-S-E isolated from callus pellets and supernatants, respectively. Importantly, the levels of the anti-inflammatory cytokine IL-10 decreased only following treatment with AYC-EVs (Figure 2e). These results suggest that the metabolite composition of AYC-EVs may exert a greater inhibitory effect on the production of anti- and pro-inflammatory cytokines in mature DCs than in AYC-P-E and AYC-S-E.

### 3.3. Treatment with AYC-EVs Attenuated the Co-Stimulatory Molecules and MHC Antigens Expressed on LPS-Treated DCs 

Mature DCs initiate and regulate adaptive immunity via the expression of surface molecules and secretion of cytokines [29]. Therefore, we investigated whether treatment with AYC-EVs impaired phenotypic maturation by inhibiting the expression of co-stimulatory molecules (CD80 and CD86) and MHC class molecules (MHC-I and MHC-II) in mature DCs. As seen in Figure 3a, the surface molecule expression of LPS-treated DCs that changed upon treatment with AYC-EVs was measured through flow cytometry. Treatment with AYC-EVs decreased the expression levels of CD80, CD86, MHC-I, and MHC-II induced by LPS-treated DCs compared to only LPS-treated DCs (mature DCs). However, these phenotypic changes after AYC treatment did not induce any significant differences in immature DCs (untreated DCs) (Figure 3b).

### 3.4. Treatment with AYC-EVs Induced an Increase in Antigen Uptake Ability and Decrease in Antigen Presenting Ability in LPS-Treated DCs

Immature DCs that have low antigen-presenting ability and high antigen uptake ability are specialized in the uptake and processing of antigens, whereas mature DCs have a high antigen-presenting ability and are able to present antigens through the expression of MHC molecules [30]. To evaluate whether the antigen uptake ability of mature DCs increased or decreased upon treatment with AYC-EVs, their ability to uptake antigens upon AYC-EV treatment in LPS-treated DCs (mature DCs) was evaluated using FITC-labeled dextran (a common endocytic tracer). We found that antigen uptake levels (Dextran^+^CD11c^+^) decreased in LPS-treated DCs (Figure 4a) and were restored upon treatment with AYC-EVs, indicating that AYC-EVs can induce semi-mature or immature states in mature DCs. We next investigated whether treatment with AYC-EVs could regulate the antigen-presenting ability of LPS-treated DCs, as described in the *Materials and methods* section. When the antigen-presenting ability of MHC-I was analyzed using OVA protein, mature DCs treated with LPS and OVA protein showed significantly increased MHC-I-mediated antigen presenting ability in OVA_257–264_ compared to that of immature DCs (OVA-treated DCs). However, the increase in MHC-I-mediated antigen-presenting ability of mature DCs was suppressed upon treatment with AYC-EVs. When the antigen-presenting ability of MHC-II was analyzed using the Eα_44–76_ peptide, an increase in MHC-II-mediated antigen presenting ability (antigen presenting levels for Eα_52–68_ peptide) in LPS-treated DCs was also suppressed upon treatment with AYC-EVs (Figure 4b). 

### 3.5. Treatment with AYC-EVs Attenuated Proliferation and Activation of CD4^+^ and CD8^+^ T Cells Induced by LPS-Treated DCs

To precisely characterize the effect of AYC-EV treatment on DCs, we next focused on how AYC-EV-treated DCs affected the proliferation and activation of T cells. Here, we performed allogenic MLR assays using DCs (non-, LPS-, LPS/AYC-EVs-treated) from a C57BL/6 background and T cells (CFSE-labeled CD4^+^ and CD8^+^ T cells) from a BALB/c background, as described in the *Materials and Methods* section. As expected, LPS-treated DCs caused an increase in CD4^+^ and CD8^+^ T cell proliferation, compared to non-treated DCs (immature DCs), when co-cultured with T cells. However, LPS-treated DCs exposed to AYC-EVs reduced the proliferation of CD4^+^ (top panel) and CD8^+^ T cells (bottom panel), compared to DCs that were only exposed to LPS (Figure 5a). Further analysis of cytokines induced by T cells when interacting with DCs revealed that LPS-treated DCs significantly enhanced Th1 (IFN-γ, IL-2, TNF-α), Th2 (IL-5), and Th17 (IL-17A) cytokine expressions in CD4^+^ and CD8^+^ T cells, while DCs co-treated with LPS and AYC-EVs induced significantly lower levels of cytokines in T cells (Figure 5b). These results indicate that AYC-EVs can inhibit the phenotypic and functional maturation of DCs.

### 3.6. Protective Effect of AYC-EV Treatment in an OVA-Induced Asthma Model

The abovementioned results regarding the functions of AYC-EVs in mature DCs motivated us to test the therapeutic efficacy of AYC-EVs in an OVA-induced asthma model. OVA-challenged mice showed an increase in airway resistance compared to normal controls after exposure to 10 and 40 mg/mL methacholine. In contrast, AYC-EV^low^ (4 mg/kg)- and AYC-EV^high^ (8 mg/kg)-treated mice showed a significant reduction in airway resistance compared with OVA-challenged mice (Figure 6a). During the determination of the inflammatory cell counts and the inflammatory cytokine analysis of BALF, the OVA-challenged mice showed an increase in the number of inflammatory cells, including those of eosinophils, APCs (including macrophages and DCs), neutrophils, and lymphocytes, and elevated levels of TNF-α, IL-4, IL-5, IL-13, eotaxin, and MUC5AC, compared with normal mice. Importantly, infiltration of APCs, eosinophils, and lymphocytes induced by the OVA challenge in BALF was suppressed upon treatment with AYC-EVs; in particular, AYC-EV treatment groups showed a strong reduction in the infiltration of APCs compared with the dexamethasone (3 mg/kg) treatment groups. However, the number of neutrophils in the AYC-EV treatment groups was similar to that in the OVA-challenged group. Unlike the AYC-EV treatment groups, dexamethasone treatment groups showed a marked reduction in the infiltration of eosinophils, neutrophils, and lymphocytes compared with the OVA-challenged groups, but not in the infiltration of APCs (Figure 6b). Interestingly, AYC-EV- and dexamethasone-treated groups exhibited a suppression of inflammatory cytokines or chemokines, such as TNF-α, IL-4, IL-5, IL-13, eotaxin, and MUC5AC, in the BALF, compared with that observed in the OVA-challenged mice. In addition, OVA-challenged mice showed significantly increased total IgE and OVA-specific IgE levels in the serum, while dexamethasone- and AYC-EV-treated mice had significantly lower levels of total IgE and OVA-specific IgE in the serum than did OVA-challenged mice (Figure 6c). As expected, OVA-challenged mice showed inflammatory cell infiltration into peribronchiolar and perivascular lesions, goblet cell hyperplasia, and mucus overproduction in the lung tissue stained with PAS. In contrast, AYC-EV and dexamethasone treatment reduced inflammatory cell infiltration, goblet cell hyperplasia, and mucus production (Figure 6d). These results suggest that AYC-EV treatment, unlike dexamethasone treatment, shows a difference in relation to the inhibition of inflammatory cell infiltration, but may provide protective effects of suppressing pathological symptoms and airway inflammatory responses against asthma.

### 3.7. Treatment with AYC-EVs Confers Immune Protection against OVA-Mediated Asthma

Next, to investigate whether AYC-EVs inhibit the OVA-induced DC maturation in asthma models, single cell suspensions of splenocytes for all groups were stained with lineage cocktails, and anti-CD11c, anti-CD80, anti-MHC-I, and anti-MHC-II antibodies, followed by the analysis of the matured splenic DCs (CD80 and MHC-I expression on lineage^-^CD11c^+^MHC-II^+^ splenic DCs) using a gating strategy for flow cytometry, as shown in Figure 7a. The OVA-challenged mice (G2) showed an increase in the number of splenic DCs and the expression levels of CD80 and MHC-I, compared with the observations in normal mice (G1). AYC-EV (8 mg/kg; G4)-treated asthma-induced mice showed a decrease in the number and expression levels of the surface molecules (CD80, MHC-I) of splenic DCs induced by the OVA challenge. Importantly, similar to the result of reducing the infiltration of APCs in BALF, AYC-EV treatment showed an improved ability to reduce the number of splenic DCs, compared with the effects of dexamethasone treatment (3 mg/kg; G3). Importantly, the ability of AYC-EVs to decrease the maturation of splenic DCs was similar to that of dexamethasone (Figure 7a,b). 

T cells, the activity and differentiation of which can be regulated by DCs, play a significant role in asthma pathogenesis, particularly Th2, Th17, or mixed Th2/Th17 cells [31]. Therefore, we next confirmed the effect of AYC-EV treatment on T-cell responses in a mouse model of OVA-induced asthma. Single cell suspensions of splenocytes for all groups were treated for 4 h in the presence and absence of PMA/ionomycin, followed by analysis of Th1 (IFN-γ^+^CD4^+^ T cells), Th2 (IL-5^+^CD4^+^ T cells), Th17 (IL-17A^+^CD4^+^ T cells), and mixed Th2/Th17 (IL-5^+^IL-17A^+^CD4^+^ T cells) cells using a gating strategy for flow cytometry, as shown in Figure 8a. AYC-EV (8 mg/kg)-treated asthma-induced mice (G4) showed decreased levels of IFN-γ^+^CD4^+^, IL-5^+^CD4^+^, IL-17A^+^CD4^+^, and IL-5^+^IL-17A^+^CD4^+^ T cells as compared to the disease-causing groups (G2) (Figure 8a). Interestingly, AYC-EV treatment groups (G4) showed dramatically increased levels of regulatory T cells (Foxp3^+^CD25^+^CD4^+^) that confer protection against allergic airway disease, compared to the disease-causing groups (G2) (Figure 8b). Interestingly, all T-cell responses regulated by AYC-EV treatment were similar to those induced by dexamethasone, which is currently used to treat inflammatory diseases.

## 4. Discussion

Although numerous studies have reported the functionalities of plant callus-derived extracts, to the best of our knowledge, no functional studies have been conducted on plant callus-derived EVs. In this study, we demonstrated that callus derived EVs can be produced using an *A. yomena* callus culture system, and that these EVs consist of 17 major metabolites. Importantly, we demonstrated that AYC-EVs can inhibit T-cell proliferation and activities associated with the etiology of various inflammatory diseases by converting the phenotype and functionalities of mature DCs into those of immature DCs. Based on these immunological characteristics, it may be feasible to use AYC-EVs as therapeutic agents for inflammatory diseases. To verify this hypothesis, we examined the therapeutic effects of AYC-EVs in an animal model of asthma. AYC-EVs injected into mouse models of asthma were found to reduce the levels of major factors (an increase in bronchoconstriction, Th2 cytokine levels, inflammatory cell infiltration, MUC5AC expression, eotaxin expression, IgE production, and lung inflammation), for developing asthma, and relieve pathological and immunological symptoms in animal models.

DCs are APCs that play a central role in initiating innate and acquired immune responses [32,33]. Pathogens or autoantigens may induce the maturation of immature DCs, thereby, directly and indirectly, regulating the activity of other immune cells [33,34]. Through the process of acquiring, processing, and presenting pathogen-specific antigens in times of infection by human pathogens (viruses, bacteria, etc.), DCs induce the activation and differentiation of naïve T cells and, consequently, induce pathogen-specific T cell responses to activate host defense against pathogens [35,36]. However, excessive induction of DC maturation leads to excessive cytokine secretion, resulting in the stimulation of normal cells, induction of cell apoptosis, and secondary stimulation of immune cells by dead cells, ultimately causing acute or chronic inflammatory diseases [37,38,39]. As is evident, DCs play an important role in regulating the host immune response and are used in research aimed at understanding disease pathogenesis or therapeutic mechanisms, in addition to drug development [35,40,41]. Interestingly, after inducing maturation of DCs using LPS, and treating these DCs with AYC-EVs, we observed that AYC-EVs inhibited not only LPS-induced phenotypic maturation by inhibiting the expression of surface molecules in DCs, increasing the internalization of extracellular antigens, and suppressing the antigen-presenting ability of DCs, but also the functional maturation of DCs by reducing the expression of pro- and anti-inflammatory cytokines. In addition, unlike DCs treated with LPS alone, LPS-treated DCs that were additionally treated with AYC-EVs inhibited T cell proliferation and various T cell activities (Th1, Th2, Th17, and CD8 T cells) when interacting with CD4^+^ and CD8^+^ T cells. These results demonstrated that treatment with AYC-EVs can inhibit pathogenic or excessive T cell responses observed during various inflammatory diseases mediated by mature DCs.

Allergic asthma, a T cell-mediated inflammatory disease, is characterized by airway hyper-responsiveness, bronchoconstriction, elevated blood IgE levels, and airway inflammation caused by genetic and environmental factors [42,43]. Allergic asthma caused by exposure to allergens is observed when DCs react to the recognition of allergens and induction of differentiation of naïve CD4^+^ T cells into Th2 and Th17 cells, thereby promoting or inducing the secretion of factors that induce asthma [43,44,45]. Th2 immunity is generally predominant in patients with mild asthma, whereas Th17, or combined Th2 and Th17, (Th2/Th17) immunity is predominant among those with severe asthma [43]. Given the immunological characteristics of allergic asthma, regulating the maturation of DCs to control Th2 and Th17 cells may significantly relieve the symptoms of asthma [44,46]. Interestingly, although the AYC-EV treatment was not sufficient to decrease the infiltration of inflammatory cells (particularly eosinophils, neutrophils, and lymphocytes) in BALF of allergic asthma (unlike dexamethasone treatment), the infiltration of APCs (including DCs and macrophages) sharply decreased in the AYC-EV treatment groups, compared with the observation in the dexamethasone treatment group. In addition, we found that the oral administration of AYC-EVs in ovalbumin-sensitized asthmatic mouse models reduced OVA-induced DC maturation, Th2, and Th2/Th17-mediated immune responses, as well as the pathological symptoms (an increase in bronchoconstriction and inflammation) of allergic asthma, and the levels of the major factors (upregulation of Th2 cytokines, IgE, MUC5AC, and eotaxin expressions in serum or BALF) inducing asthma. These results indicate that the therapeutic effect of AYC-EVs against allergic asthma may be caused by down-regulation of T cell activation via the inhibition of DC maturation and infiltration.

Many studies have demonstrated that Th1 cells, which are involved in the antagonism of Th2 and IgE synthesis, are important immune cells that alleviate the symptoms of allergic asthma [47,48]. However, recent studies have reported that Th1-cell activation can induce lung inflammation when the symptoms of asthma worsen [49]. The effect of Th1 cells on the onset and symptoms of asthma remains unclear. It is clear that Th1-mediated immune responses can aggravate various inflammatory diseases [50]. Therefore, discovery and assessment of drug candidates that can not only inhibit Th2- and Th17-mediated responses, but also Th1-mediated responses, is urgently required for the development of medications for asthma, which can manifest through various symptoms and pathogenic mechanisms. In our in vivo study, Th1 immunity was reduced in ovalbumin-sensitized asthmatic mouse models in contrast to Th2 and Th2/Th17 immunity, and oral administration of AYC-EVs further inhibited Th1 immunity. In our analysis of regulatory T cells, which can inhibit various T cell activities and alleviate asthma symptoms, we observed that while the numbers of regulatory T cells were reduced in asthmatic mouse models, they were restored to a level similar to that observed in the control group upon the administration of AYC-EVs. Similar results (suppression of activated Th1, Th2, and Th17 cells) were observed after dexamethasone treatment.

Our results suggest that the secondary metabolite composition of AYC-EVs may contribute to the inhibition of excessive T cell responses by regulating DC maturation. Interestingly, AYC-EVs include various omega-3, omega-6, omega-7, and omega-fatty acids and their derivates. The functions of omega-3 and omega-6 fatty acids that can inhibit DC maturation and the omega-positive anti-inflammatory role of omega-9 fatty acids in sepsis have been reported [51]. Furthermore, linoleoyl ethanolamide plays an important role in the anti-inflammatory action of activated macrophages, and its potential as a therapeutic agent for inflammatory skin diseases has been reported [52]. AYC-EV treatment is expected to inhibit DC maturation because AYC-EVs contain large amounts of omega fatty acids, their derivatives, and linoleoyl ethanolamide. Despite these results, our research has several limitations. First, although 17 metabolites that were expected to have various immunological functions were identified, no studies were included to show whether any of these metabolites are involved in AYC-EV-related functions; thus, more detailed studies for metabolites involved in these functions need to be conducted. In addition, it is necessary to further identify various nucleic acids (DNA, mRNA, miRNA, etc.) and proteins contained in AYC-EVs, because EVs can induce biological activity by nucleic acids and proteins, as well as metabolites. These additional studies may yield important results for finding AYC-EV components that play a dominant role in their biological activity. Second, further studies on the molecular mechanism involved in inhibiting DC maturation are required to further elucidate the pharmacological properties of AYC-EVs.

In conclusion, in this study, we confirmed that AYC-EVs, which contain 17 metabolites, inhibited DC maturation and reduced excessive T cell responses induced by DC maturation. In addition, AYC-EVs could inhibit Th2 and mixed Th2/Th17 responses associated with the etiology of asthma in asthmatic mouse models and alleviate various asthma symptoms. These results indicate that AYC-EVs may be potential drug candidates for various inflammatory diseases. Taken together, the current results can provide various advantages for the development of plant-derived therapeutic agents for the treatment of various inflammatory diseases. Callus-derived EVs can be used as interesting materials to replace plant-derived therapeutic agents, and these systems may be an alternative technology to supplement or solve various problems (plants: change in EV characteristics according to environmental changes; mammalian cells: high cost of growth medium) that arise in the development of other sources of EVs.

## Figures and Tables

**Figure 1 cells-11-02805-f001:**
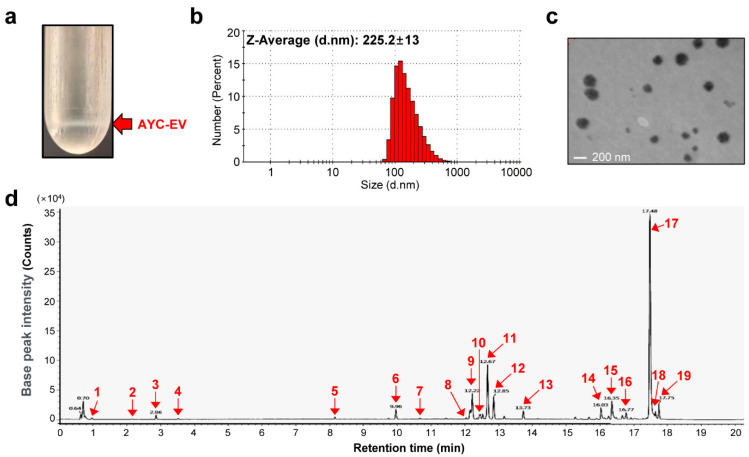
Characterization of AYC-EVs and analysis of secondary metabolites contained in AYC-EVs. (**a**) Using a TFF system and cushioned ultracentrifugation, AYC-EVs were isolated as described in the *material and methods* section. The red arrow indicates AYC-EVs separated using cushioned ultracentrifugation. (**b**) Particle sizes of isolated AYC-EVs were analyzed using DLS (n = 4). (**c**) TEM image of isolated AYC-EVs. (**d**) The metabolites contained in AYC-EVs were analyzed using Q-TOF MS in ESI-positive mode. The U PLC-QTOF/MS chromatogram indicates the following: 1 purine derivatives (guanine), 1 purine nucleosides (adenosine), 2 amino acids (L-phenylalanine, L-tryptophan), 5 fatty acyls (2-hydrolinolenic acid, 17-hydrolinolenic acid, α-linolenic acid, stearidonic acid, methyl linoleate), 3 glycerophospholipids (LysoPC(18:2), LysoPC(16:1), LysoPC(18:1)), 5 fatty amides (adrenoyl ethanolamide, linoleoyl ethanolamide, palmitic amide, 13-docosenamide, octadecanmide).

**Figure 2 cells-11-02805-f002:**
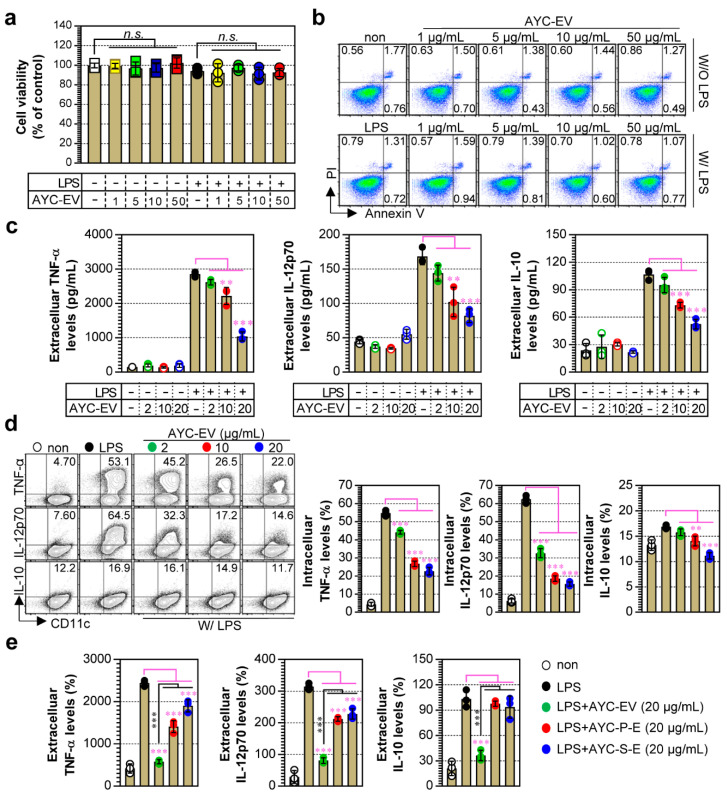
Analysis of DC viability and cellular cytotoxicity and cytokine production induced by AYC-EVs treatment in immature and mature DCs. (**a**–**c**) Immature DCs (non-stimulated DCs) and mature DCs (DCs treated with 100 ng/mL LPS for 2 h) were treated with AYC-EVs (1 to 50 μg/mL). Following incubation for 24 h, cell viability, cellular cytotoxicity, and extracellular cytokine levels (TNF-α, IL-12p70, IL-10) were measured. (**a**) Cell viability was analyzed using the EZ-Cytox kit reagent. Bar graph displays mean ± SD (n = 3 samples). n.s.: not significant (**b**) Cellular cytotoxicity of AYC-EVs was confirmed using flow cytometry (FACS) with Annexin V and PI staining. Representative FACS data from one out of three independent experiments are shown. (**c**) Extracellular cytokine levels were measured in culture supernatants using cytokine-specific ELISA kits. Bar graphs display mean ± SD (n = 3 samples). (**d**) DCs exposed to LPS for 2 h were treated with AYC-EVs (2, 10, and 20 μg/mL) in the presence of 1x Protein Transport Inhibitor Cocktail. Following incubation for 12 h, cells were immunostained for CD11c, TNF-α, IL-12p70, and IL-10, as described in the Section 2, and intracellular cytokine levels were analyzed using flow cytometry. (**e**) Mature DCs (DCs treated with 100 ng/mL LPS for 2 h) were treated with AYC-EVs, AYC-P-E, and AYC-S-E, each at a concentration of 20 μg/mL. TNF-α, IL-12p70, and IL-10 levels were measured in culture supernatants using cytokine-specific ELISA kits. Bar graphs display mean ± SD (n = 3 samples). Bar graphs display mean ± SD (n = 3 samples). All experiments were repeated three times with similar results. ** *p* < 0.01, *** *p* < 0.001. Non: untreated DCs, W/O LPS: DCs without LPS treatment, W/ LPS: DCs with LPS treatment, MFI: mean fluorescence intensity.

**Figure 3 cells-11-02805-f003:**
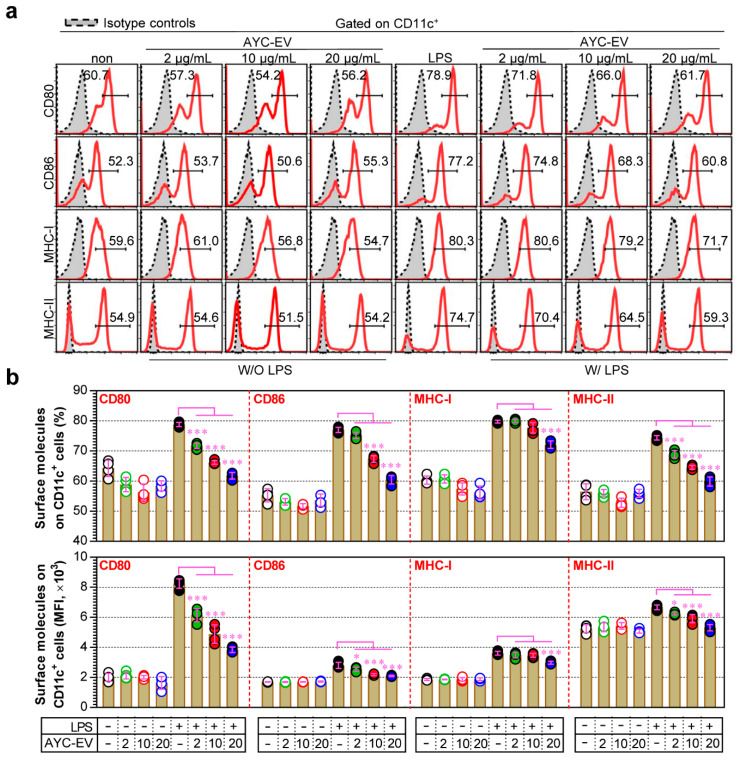
Change in expression of co-stimulatory and MHC class molecules after AYC-EVs treatment in immature and mature DCs. (**a**,**b**) Immature DCs (non-stimulated DCs) and mature DCs (DCs treated with 100 ng/mL LPS for 2 h) were treated with AYC-EVs (2, 10, and 20 μg/mL). Following incubation for 24 h, cell surface molecule expressions were examined using flow cytometry in DCs immunostained for CD11c, CD80, CD86, MHC-I, and MHC-II, as described in the Section 2. Bar graphs display mean ± SD (n = 3 samples). All experiments were repeated three times with similar results. * *p* < 0.05, *** *p* < 0.001. Non: untreated DCs, W/O LPS: DCs without LPS treatment, W/ LPS: DCs with LPS treatment, MFI: mean fluorescence intensity.

**Figure 4 cells-11-02805-f004:**
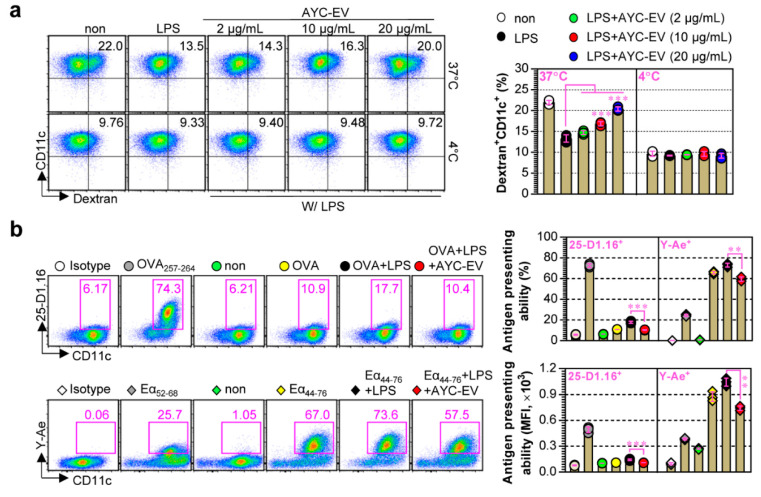
Antigen uptake and antigen presenting abilities regulated by AYC-EV treatment in DCs exposed to LPS. (**a**) Immature DCs were stimulated with LPS (100 ng/mL) or with LPS and AYC-EVs (2, 10, and 20 μg/mL). Following incubation for 24 h, cells were exposed to FITC-conjugated Dextran (0.5 mg/mL) for 30 min at 37 °C and 4 °C (negative control for antigen uptake). The cells were then stained using anti-CD11c antibody. CD11c^+^ dextran^+^ cells were detected using flow cytometry. (**b**) Antigen presenting abilities for MHC-I and MHC-II were confirmed in OVA (MHC-I)- or Eα_44–76_ peptide (MHC-II)-treated DCs as described in the *material and methods* section. OVA_257-264_ and Eα_52-68_ peptides were used as positive controls for antigen presenting of MHC-I and MHC-II. CD11c^+^25-D1.16^+^ cells indicate OVA_257–264_/MHC-I complexes. CD11c^+^Y-Ae^+^ cells indicate Eα_52-68_/MHC-II complexes. All bar graphs display mean ± SD (n = 3 samples). All experiments were repeated three times with similar results. ** *p* < 0.01, *** *p* < 0.001. Non: untreated DCs, W/O LPS: DCs without LPS treatment, W/ LPS: DCs with LPS treatment, MFI: mean fluorescence intensity.

**Figure 5 cells-11-02805-f005:**
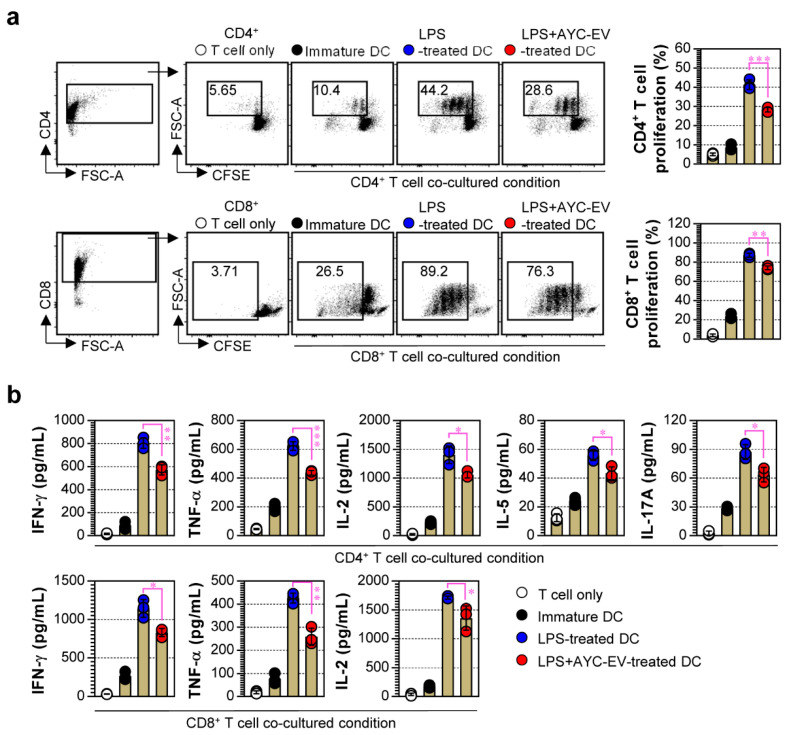
The capacity of AYC-EVs to regulate LPS-treated DCs-mediated T cell proliferation and activation. (**a**,**b**) splenic CD4^+^ and CD8^+^ T cells sorted from BALB/c mice were stained with CellTrace (CFSE), and CFSE-labeled T cells were co-cultured with DCs treated with PBS (immature DC), LPS (100 ng/mL), or LPS with AYC-EVs (20 μg/mL) in presence of anti-CD3 (1 μg/mL) and anti-CD28 (1 μg/mL). Following incubation for 48 h, cells and supernatants were harvested. (**a**) The cells were stained with anti-CD4 and anti-CD8 antibodies. Levels of T cell proliferation (CD4^+^CFSE^-^ and CD8^+^CFSE^-^ cells) were evaluated using flow cytometry. (**b**) Cytokine production in CD4^+^ and CD8^+^ T cells was measured in culture supernatants using cytokine-specific ELISA kits. All bar graphs display mean ± SD (n = 3 samples). All experiments were repeated three times with similar results. * *p* < 0.05, ** *p* < 0.01, *** *p* < 0.001.

**Figure 6 cells-11-02805-f006:**
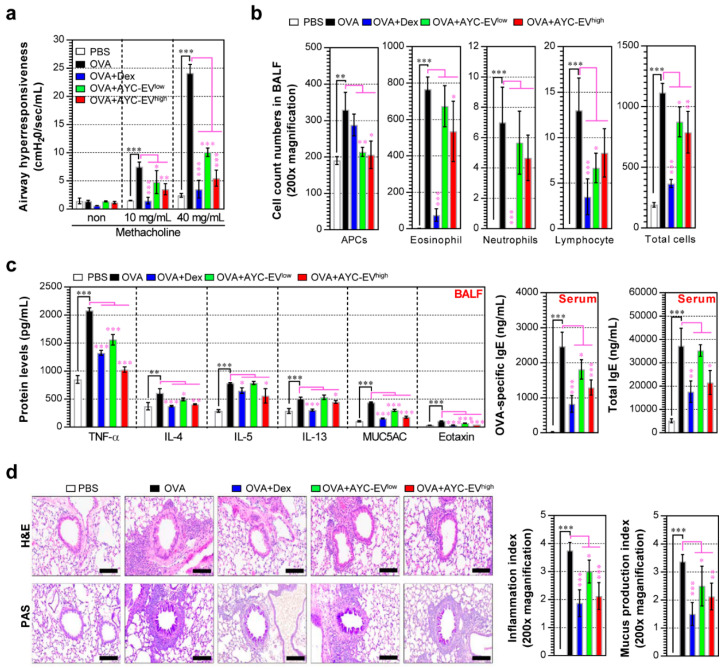
Effects of AYC-EV administration on airway hyper-responsiveness, cytokine production, and histological changes in an OVA-induced asthma model. (**a**) Airway hyper-responsiveness was evaluated using Flexivent 24 h after the last OVA challenge in an OVA-induced asthma model. (**b**) The inflammatory cells (eosinophils, APCs, neutrophils, and lymphocytes) in BALF were deposited on slides and stained with Diff-Quik stain reagent and counted in a double-blind manner in 3 areas for each slide. (**c**) The levels of TNF-α, IL-4, IL-5, IL-13, MUC5AC, and eotaxin in the BALF and the OVA-specific IgE and total IgE in the serum were determined using ELISA. (**d**) Histopathological analysis of airway inflammation and mucus production was performed in the lung tissues using H&E and PAS staining. Scale bars, 100 μm. PBS: OVA: group administrated with PBS; OVA: group administrated with OVA; OVA+Dex: group administrated with OVA and dexamethasone (3 mg/kg); OVA+AYC-EV^low^: group administrated with OVA and 4 mg/kg AYC-EVs; and OVA+AYC-EV^high^: group administrated with OVA and 8 mg/k1000g AYC-EVs. Values are presented as means ± SD (n = 7 mice). * *p* < 0.05, ** *p* < 0.01, *** *p* < 0.001. APCs: antigen-presenting cells.

**Figure 7 cells-11-02805-f007:**
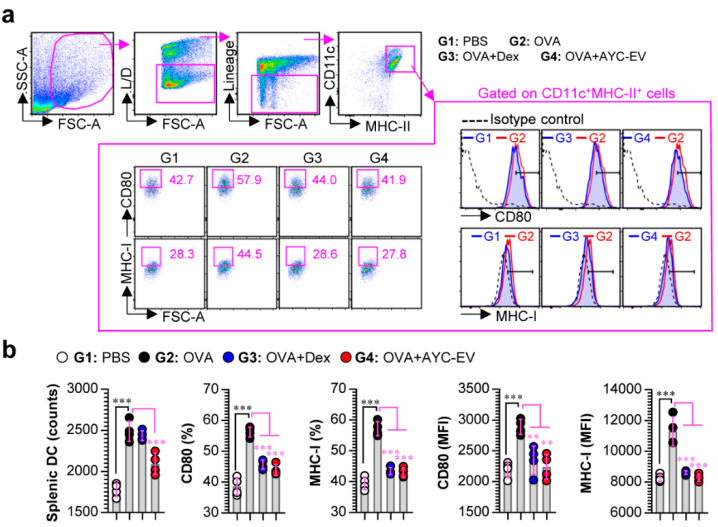
Effects of AYC-EV administration on DC maturation in an OVA-induced asthma model. (**a** and **b**) Single cell suspensions isolated from mice (at 2 days after the last OVA challenge) of PBS-, OVA-, OVA + Dex-, and OVA + AYC-EV (8 mg/kg)-administrated groups (n = 4 mice per group) were immunostained for dead cells (L/D), lineage cocktails (anti-CD3, anti-CD19, anti-NK1.1), CD11c, CD80, MHC-I, and MHC-II, as described in the materials and methods section, and cells were analyzed using flow cytometry. (**a**) Gating strategy for surface molecule (CD80, MHC-I) expression on splenic DCs (CD11c^+^MHC-II^+^ cells). (**b**) Absolute numbers and expression levels (percentages and MFI: mean fluorescence intensity) of surface molecules on splenic DCs (CD11c^+^MHC-II^+^ cells). The results of one representative study out of at least two independent studies are presented. ** *p* < 0.01, *** *p* < 0.001.

**Figure 8 cells-11-02805-f008:**
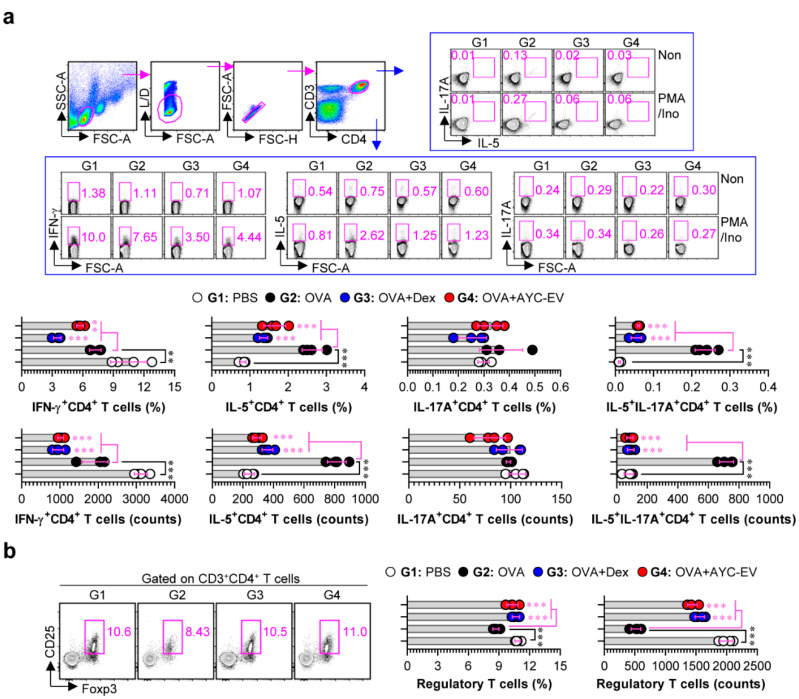
Effects of AYC-EV administration on T cell responses in an OVA-induced asthma model. (**a**) Splenic T cells isolated from mice (at 2 days after the last OVA challenge) of PBS-, OVA-, OVA + Dex-, OVA + AYC-EV (8 mg/kg)-administrated groups (n = 4 mice per group) were stimulated in vitro with Cell Stimulation Cocktail (PMA/Ino) for 4 h. Following incubation for 4 h, T cells were immunostained for dead cells (L/D), CD3, CD4, IFN-γ, IL-5, and IL-17A as described in the *material and methods* section, and percentages and absolute numbers of Th1 (CD3^+^CD4^+^IFN-γ^+^ cells), Th2 (CD3^+^CD4^+^IL-5^+^), Th17 (CD3^+^CD4^+^IL-17A^+^ cells), and mixed Th2/Th17 cells (CD3^+^CD4^+^IL-5^+^IL-17A^+^ cells) were analyzed using flow cytometry. (**b**) Splenic T cells isolated from mice from each group were immunostained for L/D, CD3, CD4, CD25, and Foxp3 as described in the *material and methods* section, and percentages and absolute numbers of regulatory T cells (CD3^+^CD4^+^CD25^+^Foxp3^+^ cells) were analyzed using flow cytometry. The results of one representative study out of at least two independent studies are presented. ** *p* < 0.01, *** *p* < 0.001.

**Table 1 cells-11-02805-t001:** Metabolites identified in AYC-EVs analyzed using ESI-positive mode in UPLC-QTOF/MS.

Peak No.	RT (min)	Identification	Exact Mass (*m*/*z*)	Fragment Ions (*m*/*z*)
1	0.96	Guanine	152.05	135, 110, 109
2	2.16	Adenosine	268.09	230, 136
3	2.86	L-Phenylalanine	166.08	149, 120, 103, 91, 77
4	3.51	L-Tryptophan	205.09	188, 170, 146, 144, 118, 115, 74
5	8.15	2-Hydroxylinolenic acid	295.22	277, 259, 109, 95
6	9.96	Adrenoyl ethanolamide	376.25	358, 348, 292
7	10.66	17-Hydroxylinolenic acid	295.21	277, 259, 219, 179, 95
8	12.03	LysoPC(18:2)	520.33	502, 184, 104
9	12.22	α-Linolenic acid	279.22	261, 109, 95, 81
10	12.46	Stearidonic acid	277.21	259, 149, 135, 121, 81
11	12.67	Methyl linoleate	295.21	277, 165, 151
12	12.85			277, 151, 81
13	13.73	Linoleoyl ethanolamide	324.28	306, 263, 81
14	16.03	LysoPC(16:1)	494.57	184
15	16.35	LysoPC(18:1)	522.59	184
16	16.77	Palmitic amide	256.25	238, 184, 83
17	17.48	13-Docosenamide	338.33	321, 303, 97
18	17.63	Octadecanamide	284.29	184, 97
19	17.75			97

## Data Availability

The datasets used and/or analyzed during the current study are available from the corresponding author on reasonable request.

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
