# Peer review of "Immunological Effects of Aster yomena Callus-Derived Extracellular Vesicles as Potential Therapeutic Agents against Allergic Asthma"

_cells, 2022, doi:10.3390/cells11182805_

Round 1
Reviewer 1 Report
The authors in the manuscript explored the immunological effect of the extracellular vesicles (EVs) derived from the callus culture of an herbal plant, Aster yomena. Using in vitro system, the authors found the suppressive activity of the EVs on the production of pro- and anti-inflammatory cytokines from murine bone marrow-derived dendritic cells differentiated into mature DCs by LPS treatment. EV treatment of mature DCs also led to a decrease in the expression of co-stimulatory and MHC molecules, antigen presentation ability, and the ability to activate T cells. The authors further explored the metabolite composition of the EVs and detected a number of lipid metabolites. In addition, EV treatment in a mouse model of ovalbumin-induced asthma alleviated a number of inflammation-related parameters. Finally, the authors found suppression of CD4+ Th cell activity from the EV-treated group of asthma-induced animals, similar to the effect seen with anti-inflammatory dexamethasone-treated group.
Overall, the study is well-designed and the authors did a thorough characterization of the immunological effect of Aster yomena-derived EVs on dendritic cells using the in vitro assay system. Studies conducted using in vivo inflammatory disease model, however, seemed incomplete. Given the striking effect of EVs on suppressing mature DC activity as detected in the in vitro study, the authors should have profiled the DC phenotypes and function in the in vivo asthma model. Following are the points that the authors need to address to make the study more comprehensive and informative.
1. DCs should be isolated from bronchoalveolar lavage fluid and lung tissues from the asthma model and profiled to assess the anti-inflammatory effect of EV treatment on DCs in a disease setting.
2. Figure 6b suggests that EV treatment even at a higher dose is not sufficient to decrease the inflammatory cell count in the BALF unlike dexamethasone treatment (especially for eosinophils and total cells). The authors should address this in the discussion, perhaps EVs do not prevent immune cell infiltration but alleviate inflammation by dampening immune cell function. The authors should also mention what other cells they have analyzed in the same figure. They may include a separate plot for the other cells to accommodate the lower numbers and clarify the significant effect of EV treatment on them.
3. The authors should use the absolute numbers of different CD4+ Th cell populations to generate the graph (Fig 7) and compare different treatment groups. This is important as the treatment itself may affect the overall proportion of T cells in splenocytes. The change in the proportion of regulatory T cells (Fig 7b) from 8.5% in the disease-causing group to 11% in the EV-treated group cannot be considered a dramatic increase (sentence 560).
Author Response
Response to Reviewer #1
Reviewer #1
Major Comments
The authors in the manuscript explored the immunological effect of the extracellular vesicles (EVs) derived from the callus culture of an herbal plant, Aster yomena. Using in vitro system, the authors found the suppressive activity of the EVs on the production of pro- and anti-inflammatory cytokines from murine bone marrow-derived dendritic cells differentiated into mature DCs by LPS treatment. EV treatment of mature DCs also led to a decrease in the expression of co-stimulatory and MHC molecules, antigen presentation ability, and the ability to activate T cells. The authors further explored the metabolite composition of the EVs and detected a number of lipid metabolites. In addition, EV treatment in a mouse model of ovalbumin-induced asthma alleviated a number of inflammation-related parameters. Finally, the authors found suppression of CD4+ Th cell activity from the EV-treated group of asthma-induced animals, similar to the effect seen with anti-inflammatory dexamethasone-treated group.
Overall, the study is well-designed and the authors did a thorough characterization of the immunological effect of Aster yomena-derived EVs on dendritic cells using the in vitro assay system. Studies conducted using in vivo inflammatory disease model, however, seemed incomplete. Given the striking effect of EVs on suppressing mature DC activity as detected in the in vitro study, the authors should have profiled the DC phenotypes and function in the in vivo asthma model. Following are the points that the authors need to address to make the study more comprehensive and informative.
We thank you for your valuable time and for providing expert comments regarding our manuscript. We have carefully considered your comments and suggestions and have attempted to address all concerns raised to the best of our abilities. We have also conducted additional experiments and added more data to the revised manuscript.
Q1. DCs should be isolated from bronchoalveolar lavage fluid and lung tissues from the asthma model and profiled to assess the anti-inflammatory effect of EV treatment on DCs in a disease setting.
[A1] We appreciate that the reviewer has raised this point. We accidentally omitted the DC results (anti-inflammatory effects of EV treatment on DCs in a disease setting) in the previous version of the manuscript. The relevant results have been added to the revised manuscript [Materials and Methods section “page 7 (lines 331–339)”, Figure 7, Results section “page 16 (lines 571–580)”, Discussion section “page 19 (line 669)”].
- Figure 6b suggests that EV treatment even at a higher dose is not sufficient to decrease the inflammatory cell count in the BALF unlike dexamethasone treatment (especially for eosinophils and total cells). The authors should address this in the discussion, perhaps EVs do not prevent immune cell infiltration but alleviate inflammation by dampening immune cell function. The authors should also mention what other cells they have analyzed in the same figure. They may include a separate plot for the other cells to accommodate the lower numbers and clarify the significant effect of EV treatment on them.
[A2] We thank the reviewer for the suggestions and comments, which have helped us to improve our manuscript further. As suggested, we have discussed the therapeutic effects of AYC-EVs in relation to immune cell functions and infiltration in the revised manuscript [Discussion section “page 19 (line 665-675)”]. Moreover, the structure of data in Figure 6b has been rephrased to clarify the significant effect of EV treatment on immune cell infiltration.
Q3. The authors should use the absolute numbers of different CD4+ Th cell populations to generate the graph (Fig 7) and compare different treatment groups. This is important as the treatment itself may affect the overall proportion of T cells in splenocytes. The change in the proportion of regulatory T cells (Fig 7b) from 8.5% in the disease-causing group to 11% in the EV-treated group cannot be considered a dramatic increase (sentence 560).
[A3] We appreciate your excellent suggestion. As indicated, absolute number data with respect to different CD4+ Th cell populations have been added to the revised manuscript [Figure 8 in the revised manuscript]. The relevant descriptions in the Materials and Methods have also been added to the revised manuscript [Materials and Methods section “page 7 (lines 357–359)”].
Reviewer 2 Report
The authors used EVs isolated from A. yomena callus (AYC-EVs) and demonstrated that AYC-EVs can inhibit DC maturation and APC function. Importantly, they found that AYC-EVs can suppress OVA-induced asthma and related phenotypes, suggesting that AYC-EVs could severe as therapeutic reagents for asthma. The study is well-designed and well-written with a lot of detailed information on Methods and Results. However, there are a couple of concerns:
1. The proper control is missing. For example, It is not clear about the strengthens of AYC-EVs as compared to other substances derived from A. yomena callus. Similarly, it is not clear whether EVs from AYC are better than those from other sources.
2. Given that AYC-EVs showed significant inhibition of DC maturation and downstream APC function and cytokine release, it is critical to find what are the major components contained in EVs that play a dominant role. Although 17 metabolites were identified, no studies were included to show whether any of those metabolites are involved in these AYC-EVs-related functions.
3. Discussion section should be precise with the major findings, strengthens, limitations, and plans for future studies. The current version seems to introduce asthma and its immunological features.
Author Response
Response to Reviewer #2
Reviewer #2
Major Comments
The authors used EVs isolated from A. yomena callus (AYC-EVs) and demonstrated that AYC-EVs can inhibit DC maturation and APC function. Importantly, they found that AYC-EVs can suppress OVA-induced asthma and related phenotypes, suggesting that AYC-EVs could severe as therapeutic reagents for asthma. The study is well-designed and well-written with a lot of detailed information on Methods and Results. However, there are a couple of concerns:
We thank you for your valuable time and for providing expert comments regarding our manuscript. We have carefully considered your comments and suggestions and have attempted to address all concerns raised to the best of our abilities. We have also conducted additional experiments and added more data to the revised manuscript.
- The proper control is missing. For example, It is not clear about the strengthens of AYC-EVs as compared to other substances derived from A. yomena callus. Similarly, it is not clear whether EVs from AYC are better than those from other sources.
[A1] We are very thankful to the reviewer for their important and thorough comments. We completely agree with the review’s comments regarding the functional analysis of AYC-EVs, as compared to other substances derived from A. yomena callus. Therefore, we have added results showing the anti-inflammatory activity of other substances (supernatant extracts, AYC-S-E, and pellet extracts, AYC-P-E) derived from A. yomena callus and compared it with the response to AYC-EVs, in accordance with the reviewer’s suggestion. The relevant results have been added to the revised manuscript [Materials and Methods section “page 3 (lines 136–146)”, Figure 2e, Results section “page 10 (lines 426–430)”, Figure 2 legend “page 11 (lines 444–447)”]. In addition, we completely agree with the reviewer’s comments regarding the functional analysis comparing AYC-EVs to EVs of other sources. In fact, EVs isolated from the A. yomena leaf were used to examine their effect on inhibiting DC maturation. Importantly, EVs (20 μg/mL) isolated from the A. yomena leaf induced anti-inflammatory effects, similar to that with EVs (20 μg/mL) derived from the A. yomena callus. However, the primary aim of our current study was to investigate the therapeutic potential of AYC-EVs by evaluating immune responses and therapeutic efficacy. In addition, we are preparing additional papers to report these results (anti-inflammatory activity of EVs isolated from the A. yomena leaf). Thus, we did not include those data in the revised manuscript.
- Given that AYC-EVs showed significant inhibition of DC maturation and downstream APC function and cytokine release, it is critical to find what are the major components contained in EVs that play a dominant role. Although 17 metabolites were identified, no studies were included to show whether any of those metabolites are involved in these AYC-EVs-related functions.
[A2] We appreciate your excellent comment. We agree with the reviewer’s opinion that the major components contained in AYC-EVs that play a dominant role in their effect should be identified. As the reviewer suggested, this is a limitation of our research. Importantly, the anti-inflammatory effect induced by AYC-EVs could be caused by nucleic acids (DNA, mRNA, miRNA, etc.) and proteins, as well as metabolites. Thus, we have added study limitations and further directions in the Discussion section of the revised manuscript to address this point [Discussion section “page 20 (lines 702–712)].
- Discussion section should be precise with the major findings, strengthens, limitations, and plans for future studies. The current version seems to introduce asthma and its immunological features.
[A3] We would like to thank the reviewer for their excellent comments and suggestions, which have been valuable in improving the quality of our manuscript. To address this concern, we have additionally discussed major findings, strengths, limitations, and plans for future studies in the discussion section [Discussion section “page 20 (lines 702–712 and lines 718–724)].
Round 2
Reviewer 1 Report
The difference among treatment groups (G1 through G4) in the expression of co-stimulatory molecule and MHCs through the histogram peaks presented in Fig 8b is not convincing in contrast to the clear difference seen on the graphs done with quantification of % positive cells. The authors should overlay the histograms among groups and also include scatter plots with % values to provide better clarity.
The authors should investigate the infiltration of DCs in the lung homogenate or BALF as also suggested in comment 1.
Additional comments:
Line 426-428: “Importantly, AYC-EVs showed better anti-inflammatory activity (effects that can reduce TNF-α, IL-12p70, and IL-10 production) in LPS-treated DCs than AYC-P-E and AYC-S-E isolated from callus pellets and supernatants (Figure 2e).”
IL-10 is known as a cytokine with anti-inflammatory properties and hence contradicts the statement. The authors should address it.
Author Response
Response to Reviewer #1
Reviewer #1
Major Comments
Q1. The difference among treatment groups (G1 through G4) in the expression of co-stimulatory molecule and MHCs through the histogram peaks presented in Fig 8b is not convincing in contrast to the clear difference seen on the graphs done with quantification of % positive cells. The authors should overlay the histograms among groups and also include scatter plots with % values to provide better clarity.
[A1] Thank you for your advice. As the reviewer pointed out, we agree that the histogram data presented in Figure 7 is not convincing. We revised the gating strategies of splenic DCs to better clarify the difference among the treatment groups for splenic DC maturation. First, because specific markers for DCs present in the spleen are CD11c and MHC-II, splenic DC populations were gated using CD11c+ and MHC-II+ cells to better identify the splenic DC population. Next, we analyzed the expression of surface molecules in the splenic DC population (CD11c+MHC-II+ cells). As suggested, we have included the overlay histogram and dot plot data (with % values) for the expression of surface molecules in splenic DCs. The relevant results have been included in the revised manuscript [Materials and Methods section “Page 7 (Lines 337–338)”, Figure 7, Results section “Page 16 (Lines 586–597)”, Figure legend sensation “Page 17 (Lines 603-606)”]
Q2. The authors should investigate the infiltration of DCs in the lung homogenate or BALF as also suggested in comment 1.
[A2] We appreciate your comments. In fact, the Diff-Quik stain reagent used in Figure 6B allows both macrophages and dendritic cells (DCs) to be stained, and upon being stained, the form and staining characteristics between the two cells are very similar. Thus, results of macrophage infiltration in BALF may include dendritic cells. We, therefore, modified the macrophages indicated in the manuscript to antigen-presenting cells (APCs), including DCs and macrophages. In order to clarify the influence of AYC-EV treatment on inflammatory cell infiltration, we have also drastically revised the content of the result interpretations that may have originally been the source of confusion. [Figure 6, Materials and Methods section “Page 7 (Lines 312–313)”, Results Section “Page 15 (Lines 542–568)”, Discussion section “Page 19 (Lines 684-696)]. We wanted to obtain accurate results of DC infiltration by FACS analysis through additional animal experiments, but we did not have enough time considering the resubmission deadline (within 10 days) to conduct additional animal experiments to include in the revised manuscript. If our interpretation of the results poses any issues, we can solve this problem through further experiments, but the time required for this will be at least three weeks.
Additional comments:
Q1. Line 426-428: “Importantly, AYC-EVs showed better anti-inflammatory activity (effects that can reduce TNF-α, IL-12p70, and IL-10 production) in LPS-treated DCs than AYC-P-E and AYC-S-E isolated from callus pellets and supernatants (Figure 2e).”
IL-10 is known as a cytokine with anti-inflammatory properties and hence contradicts the statement. The authors should address it.
[A1] Thank you so much for your critical advice. This sentence has been rephrased in the revised manuscript to clarify the originally intended meaning [Results section “Page 10 (Lines 426–433)”].

Round 3
Reviewer 1 Report
The authors addressed the raised points and improved the manuscript.